# Enhancement of Photocatalytic Activity under Visible Light Irradiation via the AgI@TCNQ Core-Shell Structure

**DOI:** 10.3390/ma12101679

**Published:** 2019-05-23

**Authors:** Wanli Xie, Li Liu, Wenquan Cui, Weijia An

**Affiliations:** Chemical Engineering, North China University of Science and Technology, Tangshan 063009, China; xwl199810@163.com (W.X.); chemll@126.com (L.L.); wkcui@163.com (W.C.)

**Keywords:** TCNQ, core-shell structure, photocatalytic activity

## Abstract

In this paper, a AgI@TCNQ photocatalyst with a core-shell structure was reported. A two-dimensional TCNQ (7,7,8,8-Tetracyanoquinodimethane) nanosheet, with a π-π conjugate structure, was used as a shell layer to realize the flexible coating on the surface of AgI nanoparticles. These special core-shell structure composites solve the key problems of the small interface of the bulk composites and the lesser charge transfer paths, which could accelerate the migration of photogenerated carriers. Thus, the AgI@TCNQ photocatalysts showed the better photodegradation performance for the methylene blue (MB) solution, and the degradation rate of AgI@TCNQ (1 wt.%) composite was 1.8 times than AgI under irradiation. The reactive species trapping experiments demonstrated that ·O_2_^−^, h^+^, and ·OH all participated in the MB degradation process. The photocatalytic mechanism of AgI@TCNQ composites could be rationally explained by considering the Z-scheme structure, resulting in a higher redox potential and more efficient separation of charge carriers. At the same time, the unique core-shell structure provides a larger contact area, expands the charge transport channel, and increases the surface active sites, which are beneficial for improving photocatalytic performance.

## 1. Introduction

Photocatalytic technology can directly utilize sunlight to achieve deep degradation of organic pollutants; has the advantages of strong oxidizing ability, low cost, and no pollution; and has received widespread attention [1,2]. Photocatalysts, as the key to photocatalysis technology, have always been a research hotspot. Recently, the AgI photocatalyst has exhibited good photocatalytic degradation activity, and it could be further improved by regulating the morphology of AgI [3,4,5]. However, AgI is prone to photo-corrosion and agglomeration; this decreases the active sites and increases the recombination of photogenerated carriers, which is not conducive to photocatalytic performance [6]. In order to solve the problem of photo-generated charge easy to recombine and the poor stability of AgI, constructing heterostructures could promote the separation of photogenerated charger carriers through the matched energy level structure, which avoids the restoration of AgI when exposed to light. Many studies, such as the AgI-Ag_3_PO_4_ [7], AgI/BiVO_4_ [8], and AgI/RGO [9], have confirmed that this modification method could improve the separation efficiency of photogenerated electron hole pairs and enhance photocatalytic activity. However, this bulk composite has fewer interfacial contact sites, and the smaller photo-generated charge transport path inhibits the separation of charges [10]. Thus, it is very important to seek a new way to accelerate the separation ability of photo-generated charges and increase the active sites. 

The core-shell structure has obvious advantages in terms of increasing the contact area between materials due to its unique face-to-face cladding structure; accordingly, the photo-generated charge could be transferred quickly between the semiconductor materials [11,12]. Meanwhile, the shell material could also protect the core particles to some extent and prevent its direct contact with the external environment [13]. Therefore, the construction of AgI-based core-shell structure composites is an effective way to improve photocatalytic activity. It will not only solve the key problem that the phase contact brings to the small contact area, but avoid the photo-corrosion phenomenon of AgI under illumination, which can greatly promote its photocatalytic performance.

Recently, the π-π conjugated structure composite has been demonstrated to have the ability to separate the photogenerated charges in the photocatalytic charge transfer process [14,15]. The π-electrons in the π-π conjugated structure are more active and easily escape from the orbit to form free electrons, which provide an effective channel for carrier transfer [16]. Graphene [17,18,19], g-C_3_N_4_ [20,21,22], PANI [23,24], etc., are typical π-π conjugated materials, have the high charge mobility, and have been brilliantly used in the design of photocatalytic materials. Besides, TCNQ which is a typical material with a π-π conjugate structure, not only has good chemical stability and thermal stability, but also has a high charge mobility, which can promote the rapid separation of charges during electron transfer [25]. Therefore, when it is combined with other semiconductors to construct photocatalytic material, a chemical bond can be formed between the π-π conjugated material and the catalyst, which promotes rapid separation of photogenerated charges, suppresses photo-corrosion, and improves photocatalytic performance, as has been confirmed by previous works [26,27,28]. 

Based on the above analysis, we designed the composite material using TCNQ coating on the surface of AgI to build core-shell photocatalytic composites. The flexible coating of AgI with TCNQ solves the key problem of less interfacial contact sites in the traditional AgI composite system and the smaller photo-induced charge transport path to suppress charge separation. Compared with the existing core-shell photocatalyst, it has more charge transport channels and faster charge separation efficiency, thus avoiding the problem of AgI photo-corrosion. It is envisaged it will accelerate the migration of the photogenerated electron hole pair by π-π conjugation and the strong interaction of core-shell materials, and finally improve photocatalytic degradation performance.

## 2. Experimental

### 2.1. Synthesis of Photocatalysts

AgI was prepared using the precipitation method. Firstly, 5 mmol AgNO_3_ and 5 mmol KI were dissolved in 25 mL distilled water, respectively. Then, 0.25 g citric acid and polyvinylpyrrolidone (PVP) were added to the AgNO_3_ solution. After stirring for 30 min, the KI solution was added dropwise to the AgNO_3_ solution, and the final solution was further stirred for 1 h, centrifuged, washed, and dried to obtain the AgI sample. 

The synthesis steps of AgI@TCNQ were similar to our previous report [28]: dissolve an appropriate amount of TCNQ into tetrahydrofuran, prepare the TCNQ solution with a concentration of 0.005 g/mL, then take out 0.2 mL of TCNQ solution and further dilute to 100 mL tetrahydrofuran. Ultrasonic solution was sonicated for 5 h to obtain a two-dimensional TCNQ solution with a fixed concentration. Subsequently, AgI was added to the above solution ultrasound for 2 h and stirred for 24 h to obtain AgI@TCNQ composite photocatalyst. Different ratios of AgI@TCNQ composites were prepared by adjusting the amount of AgI added. M-AgI/TCNQ are mechanically mixed samples.

### 2.2. Characterization

X-ray diffraction (XRD, Rigaku Corporation, Tokyo, Japan) analysis of the sample was performed on a D/MAX 2500PC diffractometer with a Cu Ka target operating at 40 kV and an operating current of 100 mA. The morphology of the samples was observed using the S-4800 field emission scanning electron microscope (SEM, Hitachi, Tokyo, Japan), and the morphology and structure of the catalyst were observed by transmission electron microscope (TEM, JEOL Ltd., Akishima, Japan, JEM-2010). The absorption band edge of the solid matter was measured by an ultraviolet-visible spectrophotometer (Puxi, UV1901, Beijing, China), and the barium sulfate powder was used as a reference. Fluorescence spectroscopic analysis of catalysts was undertaken using a molecular fluorescence spectrometer (PL, Hitachi F-7000, Tokyo, Japan). The photocurrent and alternating current impedance (EIS) of the composite were tested using an electrochemical measuring instrument (CHI-660E, Chen Hua Instruments, Shanghai, China).

### 2.3. Photocatalytic Activity

The photodegradation performance of the catalyst was investigated by degrade MB under visible light. The light source was a 400 W Xenon lamp with a 420 nm filter outside the photoreactor to filter out the UV light. The catalyst and the degradation solution were stirred for a period of time to achieve adsorption equilibrium prior to the photodegradation experiment. During the photocatalytic degradation reaction, 3 mL of sample was taken every 10 min. The collected supernatant solutions were analyzed by the spectrometer at the characteristic absorption peak of 664 nm. 

## 3. Results and Discussion

Figure 1 shows the XRD patterns of the as-prepared photocatalysts. It is observed that the characteristic diffraction peaks of TCNQ were sharp and narrow, meaning there was high crystallinity of TCNQ. As for the pure AgI, it displayed sharp diffraction peaks at 2θ angles of 23.7°, 39.1°, and 46.3°, attributed to (111), (220), and (311) crystal planes, indexed to the mono-clinic β phase of AgI (JCPDS 09-0399) [29]. Meanwhile, the XRD patterns of the AgI@TCNQ composite were similar to the pure AgI, indicating that the introduction of TCNQ did not change the crystallinity of AgI. At the same time, there were no characteristic diffraction peaks belonging to TCNQ, which were caused by the low content of TCNQ in the composites. 

The morphology of TCNQ after sonication at 200 nm scale observed by SEM, as seen in Figure 2a. It can be observed that the sonicated TCNQ monomer exhibits a large sheet-like structure with good flexibility. However, the pure TCNQ monomer morphology exhibits a large block structure before being treated, as seen in the insert Figure 2a. The morphology of AgI is relatively regular, and the average particle size is around 200–300 nm (Figure 2b). In Figure 2c, it can be seen the AgI composites are encapsulated by flexible two-dimensional TCNQ nanosheets to form core-shell structure. At the same time, the stability of the AgI morphology is maintained. Figure 2d shows the TEM image of the AgI@TCNQ core-shell structure; there is a thin layer of TCNQ wrapped on the outer surface of AgI, with larger contact area and more reaction sites, which solves the problem of the small interface and surface active sites of the body contact. In addition, this structure provides more channels for the migration of the photo-generated charge, which is conducive to the improvement of photocatalytic activity [30].

The UV-visible diffuse reflectance pattern of TCNQ, AgI, and AgI@TCNQ samples are shown in Figure 3. The pure AgI has strong absorption capacity to visible light below 460 nm, which is consistent with the previous report [31]. Compared with AgI, the AgI@TCNQ composite exhibits slightly red-shift; meanwhile, two characteristic peaks appear in the composites, corresponding to the absorption wage of AgI at 460 nm and absorption peak of TCNQ, respectively, indicating that TCNQ could improve the absorption and utilization of visible light. The characteristic peak appearing at 460–560 nm may be caused by the interaction between AgI and TCNQ. 

Figure 4 shows the photoluminescence spectra of the prepared AgI and AgI@TCNQ; as we all know, the lower the fluorescence intensity, the lower of the photo-generated electron-hole pair recombination rate [32]. As seen from Figure 4, the pure AgI has a strong emission peak at 310 nm, due to the recombination of the conduction band electrons with the valence band holes after excitation by AgI. When the surface of AgI was coated with TCNQ, the intensity of fluorescence spectrum was significantly weakened, and the content of TCNQ has an important effect on fluorescence intensity. When the complex amount is 1 wt.%, the fluorescence intensity reaches its lowest level, indicating that the relaxation of a fraction of AgI@TCNQ excitons may occur via charge transfer of electrons and holes rather than radiative paths [27]. Thus, the lower recombination probability of photogenerated charge carriers for AgI@TCNQ compared to pure AgI can be inferred.

Figure 5 shows the photocurrent test results of the AgI and AgI@TCNQ composites. It is widely accepted that photocurrent intensity is proportional to the photo-generated charge separation efficiency [33]. As seen from Figure 5, the composite has almost no current response under dark conditions and produces a stable photocurrent when exposed to visible light. Moreover, the photocurrent was enhanced along with the increase of TCNQ content. The photocurrent of 1 wt.% AgI@TCNQ was about 7 and 2.3 times as high as that of the pure AgI and TCNQ electrode, indicating that the separation and transition efficiency of photoinduced electrons and holes was improved via the interaction between AgI and TCNQ. The matching energy level between them and core-shell structure provided more charge transfer channels; all the aforementioned factors were advantageous for the separation of photogenerated carriers.

Figure 6 shows the electrochemical impedance spectra (EIS) of AgI and AgI@TCNQ photocatalysts. It can be clearly seen that the diameter of the Nyquist semicircle of AgI@TCNQ composite was significantly smaller, meaning the coating of the TCNQ nanosheet as a shell on the AgI surface could effectively reduce the interfacial resistance of the AgI@TCNQ system. Similarly, the 1 wt.% AgI@TCNQ composites exhibit the smallest radius, which indicates that the charge transfer resistance is the smallest and the surface electrode reaction rate is the largest [34]. All these results of EIS, photocurrent, and PL are consistent, and they powerfully proved that AgI@TCNQ system could accelerate the separation efficiency of the photo-generated charges due to the core-shell structure, thus facilitating the charge transfer, which was thought to be positive to improve the photocatalytic activity.

In photocatalytic activity test, MB solution was used as target degradation agent, and the degradation performance of AgI, TCNQ, AgI@TCNQ, and the mechanical mixed sample M-AgI@TCNQ under visible light irradiation was investigated. The blank and control experiments were set up to compare the photocatalytic activity under different conditions. As shown in Figure 7, under the dark reaction conditions, AgI@TCNQ has almost no adsorption or degradation properties to MB. Meanwhile, self-degradation of MB was basically negligible in the absence of catalyst light. When the light was turned on for photocatalytic reaction, AgI photocatalysts degraded 80% of MB in 60 min, TCNQ degrading rate of MB was 65%, and the photocatalytic activity of the mechanically mixed catalyst M-AgI@TCNQ was slightly improved compared to pure AgI, while the AgI@TCNQ (1 wt.%) composite had the best photodegradation activity, which could degrade approximately 100% MB within 30 min. The increase of composite activity in the experimental results means that the increase in activity was not due to the addition of TCNQ, but to the composite mode between the semiconductor photocatalyst. The special core-shell structure creates interactions between the materials, increases the charge transport channel, and greatly improves the separation efficiency and migration rate of the interface carrier, which is consistent with the photocurrent and EIS impedance experiments.

Figure 8 shows the first-order degradation rate constants of MB for different coverage ratios of AgI@TCNQ composites. When the surface coating amount of TCNQ is 0 wt.%, 0.3 wt.%, 0.5 wt.%, 1 wt.%, 2 wt.%, and 3 wt.%, the corresponding degradation rate constants are 0.027 min^−1^, 0.031 min^−1^, 0.036 min^−1^, 0.065 min^−1^, 0.048 min^−1^, and 0.039 min^−1^. It is not difficult to find that the content of TCNQ has an important influence on the photodegradation performance of the composite. When the TCNQ content is relatively low (<1 wt.%), the contact area between AgI and TCNQ increases with the amount of TCNQ modification. Therefore, the pathway of photogenerated charge separation increases, and the photo-generated charge carriers migration efficiency increases, effectively inhibiting the occurrence of AgI photo-corrosion, thus enhancing the photodegradation performance of the composite. Conversely, when the amount of TCNQ modification is relatively high (> 1 wt.%), the excess TCNQ forms a completely closed shell structure on the surface of the AgI; this inhibits the transfer of surface excess photoelectrons to TCNQ by AgI, caused the recombination of photogenerated charge carriers. These results confirm that the appropriate thickness of TCNQ shell could greatly influence the improvement of photocatalytic activity.

The stability of the catalyst is one of the important factors determining its practicability. Figure 9 shows the test results of the photostability of AgI and AgI@TCNQ composites. As shown, the degradation rate of MB by monomeric AgI decreased from 72% to 52% after five cycles of use. In contrast, the degradation rate of AgI@TCNQ (1 wt.%) composite photocatalyst was as high as 93% after five cycles of recycling, demonstrating that the AgI@TCNQ composites exhibited better recycling performance, which could act as a potential photocatalyst for water pollution. 

Phenol, a typical colorless organic pollutant, was selected as another contaminant to further investigate the activity of the as-prepared photocatalyst under the same conditions. As seen in Figure 10, from the apparent rate constant k of TCNQ, AgI, and AgI@TCNQ(1 wt.%), for 0.01306, 0.05121, and 0.09162 h^−1^, respectively, it is obvious that the AgI@TCNQ(1 wt.%) composites exhibit higher catalytic degradation activities for phenol solution than pure AgI and TCNQ. The apparent rate constant k is almost 1.79 and 7.01 times as high as that of pure AgI and TCNQ, which is ascribed to the fact that the TCNQ is beneficial for charge separation of the AgI@TCNQ photocatalysts, thus improving its photocatalytic activity.

In the process of photocatalytic treatment of organic pollutants, active species such as free radicals and photogenerated holes play a practical role. In order to investigate intermediate active species in photodegradation reaction, isopropanol (IPA) and EDTA-2Na were used as quenchers for hydroxyl radicals (·OH) and holes (h^+^), benzoquinone (BQ) and N_2_ were used as quenchers for superoxide radicals (·O_2_^−^) [35]. As shown in Figure 11, after introduction of N_2_ into the AgI@TCNQ (1 wt.%) reaction system, the photodegradation activity drops sharply, implying the ·O_2_^−^ plays an important role in photodegradation process. BQ is also used as a quencher for superoxide radicals. It can be seen that the degradation activity is significantly reduced, further indicating that superoxide radicals are active species in the degradation process. After adding IPA and EDTA-2Na to the reaction system, the apparent reaction rate constant k dropped from 0.065 min^−1^ to 0.0033 min^−1^ and 0.021 min^−1^, indicating both ·OH and holes are active substances produced during photodegradation process. Therefore, we can conclude that both ·O_2_^−^, h^+^ and ·OH have some effect on degradation performance, among them, ·O_2_^−^ and h^+^ play a decisive role during the photodegradation process, while ·OH has a relatively small effect on photocatalytic activity. 

As a hybrid semiconductor, the energy level structure of AgI and TCNQ has an important influence on the excitation and transfer of electrons and the recombination of carriers, which were the decisive factors of photocatalytic activity. According to the previous work of our group, combined with the literature reports [28,36], we can determine the energy level of the AgI and TCNQ, the valence band (VB), and conduction band (CB) of AgI is 2.38 and −0.42 eV; while the highest mccupied Molecular (HOMO) and lowest unoccupied molecular (LUMO) of TCNQ are 4.03 and 1.73 eV, respectively. Both AgI and TCNQ could be excited when exposed to visible light, and the existence of a small amount of Ag^+^ could reduce them to Ag. According to the migration path of photogenerated carriers, two possible mechanisms for charge transfer and migration were proposed, as seen in Scheme 1.

The first mechanism is the traditional charge separation model, as seen from Scheme 1a. From the view of band position, the excited electrons on the AgI CB will migrate to Ag nanoparticles and then be transferred to the LUMO of TCNQ. While the holes on TCNQ could migrate to AgI, such a separation mode could greatly inhibit recombination of photogenerated electron-hole pairs. However, the LUMO potential (1.73 eV) of TCNQ was more positive than the O_2_/·O_2_^−^ potential (−0.33 eV vs. NHE) [37,38], meaning that the enriched photogenerated electrons in the TCNQ LUMO could not reduce O_2_ into ·O_2_^−^. While holes that stay on the AgI valence band face the same problem, the VB potential (+2.38 eV) of AgI was more negative than H_2_O/·OH (+2.72 eV) [39,40], preventing oxidation of H_2_O molecules to ·OH. That is, the photocatalytic degradation process was only caused by the direct oxidation of pollutants by holes, which did not correspond to quenching experiment results, indicating that this charge transfer pathway may be unreasonable. 

Therefore, we propose another photocarrier transfer pathway in the AgI@TCNQ composites based on the Z-scheme mechanism (Scheme 1b). On the one hand, the electrons on the TCNQ could fast transfer to Ag nanoparticles due to the fact that the Fermi level of Ag was more positive than the potential of TCNQ LUMO. On the other hand, holes on AgI VB could be transferred to the Ag nanoparticles, where metal Ag acts as a recombination center for photogenerated carriers. Although this charge transfer method sacrifices a pair of photogenerated electron hole pairs, it ensures a stronger oxidation of AgI@TCNQ system. That is, the electrons remaining on the CB of AgI have a stronger reducing ability, which could reduce the adsorbed O_2_ into ·O_2_^−^. Meanwhile, holes accumulated in the HOMO of TCNQ have stronger oxidizing power (4.03 eV), which could react with OH^−^ to generate ·OH for the ·OH/OH^−^ potential is 2.40 eV; meanwhile, the extremely strong oxidizing ability of the holes could directly achieve the degradation of organic pollution. All these results correspond to quenching experiments. In summary, the Z-scheme system not only accelerates charge separation but also provides stronger redox capacity and further enhances photodegradation activity.

## 4. Conclusions

In summary, the Z-scheme AgI@TCNQ core-shell system was successfully fabricated using the TCNQ coating on the surface of AgI; the introduction of TCNQ greatly improved the photocatalytic activity, and the AgI@TCNQ (1 wt.%) composite degraded nearly 100% MB after 30 min under visible light irradiation. After five cycles of recycling, the photodegradation activity was still as high as 90%. The enhancement of the photodegradation activity was due to the Z-scheme structure, resulting in a higher redox potential and more efficient separation of charge carriers; meanwhile, the core-shell structure provided reactive sites and more charge transport channels, all of which were beneficial for photocatalytic activity. In summary, AgI@TCNQ core-shell photocatalyst has important application prospects in the photocatalytic treatment of organic pollutants.

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
