# Peer review of "Enhancement of Photocatalytic Activity under Visible Light Irradiation via the AgI@TCNQ Core-Shell Structure"

_materials, 2019, doi:10.3390/ma12101679_

Reviewer 1 Report

The manuscript on AgI@TCNQ is interesting and worth to be published. However, there are many important points, which must be addressed (improved) before possible publication, as shown below:

1) The authors suggested reduction of silver, based on DRS spectra - any other proof? What about XRD results? Moreover, authors wrote that “as the TCNQ content increases, the intensity of the Ag plasmon resonance absorption peak decreases” – which is not true, it is clear that the LSPR peak for silver at ca. 490 (for small, spherical silver NPs; not at “500-700 nm” as suggested by authors) was the most intensive for the largest content of TCNQ. Additionally, the sentence on the new absorption peak at 460 nm, due to AgI (in the composite) in comparison with that by AgI (without this peak) is strange – i) no absorption at 460 nm, ii) absorption at ca. 480-490 nm, due to zero-valent silver 

2) I am not sure what authors mean by using N2 as scavenger. How experiments were performed – deaerated conditions or not – if yes, then obviously no oxidative species would be formed (and the worst activity; N2 also as hole and hydroxyl radical scavenger).

3) The activity under visible light irradiation should not be tested for dyes, due to possible sensitization mechanism. It is highly recommended to test vis activity for colorless compound (e.g., phenol).

4) The lack of reference experiments is surprising, e.g., Figs. 4, 5 and 6 should include also results for TCNQ, and then the discussion on improved activity could be drawn.

5) Impedance data should be presented (and discussed) for same scales at X and Y axes.

6) There are many mistakes and errors, e.g.,

i) Grammar mistakes “was report” – “was reported”,

ii) Unnecessary use of capitals (many cases), e.g., “by Electrochemical..”,

Author Response

Dear Editors and Reviewers:

Thank you for your letter and for the reviewers’ comments concerning our manuscript entitled “Enhancement of photocatalytic activity under visible light irradiation via the AgI@TCNQ core-shell structure”.

Those comments are all valuable and very helpful for revising and improving our paper, and also very important for guidance to our research. We have studied the comments carefully and performed many additional experiments. We hope that the revision which we have made can be met with approval. Revised portions are marked in RED in the manuscript. The corrections in the paper and the responses, point by point, to the reviewers’ comments are as follows:

Response to the reviewers’ comments:

The manuscript on AgI@TCNQ is interesting and worth to be published. However, there are many important points, which must be addressed (improved) before possible publication, as shown below:

[1] The authors suggested reduction of silver, based on DRS spectra - any other proof? What about XRD results? Moreover, authors wrote that “as the TCNQ content increases, the intensity of the Ag plasmon resonance absorption peak decreases” – which is not true, it is clear that the LSPR peak for silver at ca. 490 (for small, spherical silver NPs; not at “500-700 nm” as suggested by authors) was the most intensive for the largest content of TCNQ. Additionally, the sentence on the new absorption peak at 460 nm, due to AgI (in the composite) in comparison with that by AgI (without this peak) is strange – i) no absorption at 460 nm, ii) absorption at ca. 480-490 nm, due to zero-valent silver.

Response: Thank you for the useful comment. In the AgI@TCNQ core shell system, TCNQ is coated on the surface of AgI in the form of a film. Therefore, the system mainly exists in the presence of AgI, but inevitably produces a small amount of Ag particles. Meanwhile, no peaks attributed to Ag were detected in the XRD, due to the small quantity of Ag, in agreement with previous reports[1-3]. In addition, we retested the visible light absorption performance of these samples, and the results were similar to the previous. Therefore, we have modified the inappropriate expressions in this section. The detailed discussion has been added to the revised manuscript. Thanks again to the questions raised by the reviewers, which will be of great help to the improvement of our paper.

[1] Hanbo Yu, Binbin Huang, Hou Wang, Xingzhong Yuan, Longbo Jiang, Zhibin Wu, Jin Zhang, Guangming Zeng. Facile construction of novel direct solid-state Z-scheme AgI/BiOBr photocatalysts for highly effective removal of ciprofloxacin under visible light exposure: Mineralization efficiency and mechanisms, Journal of Colloid and Interface Science, 2018, 522, 82-94.

[2] Wenjing Xue, Zhiwei Peng, Danlian Huang, Guangming Zeng, Xiaoju Wen, Rui Deng, Yang Yang, Xuelei Yan. In situ synthesis of visible-light-driven Z-scheme AgI/Bi2WO6 heterojunction photocatalysts with enhanced photocatalytic activity, Ceramics International, 2019, 45, 6340-6349.

[3] Jialiang Liang, Fuyang Liu, Jun Deng, Mian Li, Meiping Tong. Efficient bacterial inactivation with Z-scheme AgI/Bi2MoO6 under visible light irradiation, Water Research, 2017, 123, 632-641.

[2] I am not sure what authors mean by using N2 as scavenger. How experiments were performed-deaerated conditions or not-if yes, then obviously no oxidative species would be formed (and the worst activity; N2 also as hole and hydroxyl radical scavenger).

Response: Thank you for the comment. In this experiment, the reaction was carried out in a closed vessel, and N2 was continuously introduced to drive off the O2 in the system, thereby suppressing the generation of superoxide radicals. In addition, we also selected BQ as a quencher for superoxide radicals. The experimental results are consistent with the results obtained after N2, as shown in Figure 11. The detailed discussion has been added to the revised manuscript, thank you.

[3] The activity under visible light irradiation should not be tested for dyes, due to possible sensitization mechanism. It is highly recommended to test vis activity for colorless compound (e.g., phenol).

Response: Thank you for the useful comment. According to the reviewer's suggestion, we selected phenol as another target degradant, performed the HPLC test, discussed the data and detailed explanation in the revised manuscript (Fig. 10). Thanks again.

[4] The lack of reference experiments is surprising, e.g., Figs. 4, 5 and 6 should include also results for TCNQ, and then the discussion on improved activity could be drawn.

Response: We agree with the reviewer’s comment. According to the reviewer's suggestion, we performed fluorescence, photocurrent, and impedance tests on AgI, TCNQ, and AgI@TCNQ composites with different TCNQ mass fractions. The results show that when TCNQ is coated on the surface of AgI, it will promote the separation of photogenerated carriers, which is mainly due to the fact that the core-shell structure of the composite can provide more charge transfer channels to accelerate the rapid separation of charge. The results also show that the TCNQ content has an important effect on the photogenerated charge separation efficiency. 1 wt.% AgI@TCNQ exhibits the optimal charge separation efficiency, which is proportional to its photocatalytic activity, further indicating that the charge separation efficiency directly affects its photocatalytic activity. However, it should be noted that the photoelectric performance TCNQ is better than AgI, but its photocatalytic activity is lower than AgI, due to the photocurrent is a long-term process and is limited by the carrier mobility, the continuous enhancement in photocurrent can be attributed to the relatively higher carrier mobility of TCNQ than AgI[4]. In addition, the detailed discussion have been added in the revised manuscript, thanks.  

[4] Mo Zhang, Wenqing Yao, Yanhui Lv, Xiaojuan Bai, Yanfang Liu, Wenjun Jiang, Yongfa Zhu. Enhancement of mineralization ability of C3N4 via a lower valence position by a tetracyanoquinodimethane organic semiconductor, J. Mater. Chem. A, 2014, 2, 11432-11438.

[5] Impedance data should be presented (and discussed) for same scales at X and Y axes.

Response: Thank you for the comment. We have presented the Impedance data for same scales at X and Y axes, and the corresponding discussion section have added in the revised manuscript, thanks.

[6] There are many mistakes and errors, e.g.,

i) Grammar mistakes “was report” – “was reported”,

ii) Unnecessary use of capitals (many cases), e.g., “by Electrochemical..”,

Response: Thanks for point this out, and we have revised it in the manuscript. Besides, the language has been polished by one English speaking expert in the revised version. thanks.

We have tried our best to improve the manuscript and have made quite adequate changes in the revised version. We appreciate the Editors/Reviewers’ work earnestly, and hope that the correction will meet with approval.

Once again, thank you very much for your comments and suggestions.

Yours sincerely,

Weijia An

E-mail: anweijia@ncst.edu.cn

Reviewer 2 Report

The authors prepared AgI particles coated with TCNQ for enhanced stability of AgI (against photocorrosion) and enhanced photoactivity due to the role of TCNQ as a good electron conductor.  The introduction of TCNQ caused changes in the UV-vis and photoluminescence and transient photocurrent responses, and finally photodegradation rates. Their results seem to agree well with what they propose.I have a few more comments

1) The Fig 8 shows TCNQ loading dependent changes in photodegradation rates showing a peak rate at 1 wt%. Other data such as PL can be obtained and shown to see whether they also agree well with each other. 

2) After all, the authors propose that (in scheme 1) Z scheme is more probable than the tranditional pathway. Thus, the electron@ TCNQ combines with holes@AgI. Then, the role of TCNQ as a good electron conductor is not important as stressed in the abstract. The role of TCNQ and AgI need to be redefined in the abstract again.

3) The authors argue that the coating of TCNQ layer with AgI can enhance the surface active sites in the abstract. Does the active site means  the site for the oxidation by the holes?

4) The authors propose that the coating of TCNQ by AgI can enhance the stability of AgI, but the best sample AgI@TCNQ was 1wt%, which may expose both TCNQ and AgI to the solution for oxidation and reduction reactions. Thus, the enhanced stability of AgI@TCNQ vs. AgI can be quite limited or may be meaningless. This argument needs to be revised.

5) Some terminologies need to be revised to comply with common definitions in research communiities. In the abstract, "degrade rate" -> "degradation rate", "degradation ability" may need to be replaced by another proper terminology. 

"was still as high as 90%." -> 90% of what....

Author Response

Dear Editors and Reviewers:

Thank you for your letter and for the reviewers’ comments concerning our manuscript entitled “Enhancement of photocatalytic activity under visible light irradiation via the AgI@TCNQ core-shell structure”.

Those comments are all valuable and very helpful for revising and improving our paper, and also very important for guidance to our research. We have studied the comments carefully and performed many additional experiments. We hope that the revision which we have made can be met with approval. Revised portions are marked in RED in the manuscript. The corrections in the paper and the responses, point by point, to the reviewers’ comments are as follows:

Response to the reviewers’ comments:

The authors prepared AgI particles coated with TCNQ for enhanced stability of AgI (against photocorrosion) and enhanced photoactivity due to the role of TCNQ as a good electron conductor. The introduction of TCNQ caused changes in the UV-vis and photoluminescence and transient photocurrent responses, and finally photodegradation rates. Their results seem to agree well with what they propose. I have a few more comments:

[1] The Fig 8 shows TCNQ loading dependent changes in photodegradation rates showing a peak rate at 1 wt%. Other data such as PL can be obtained and shown to see whether they also agree well with each other.

Response: We agree with the reviewer’s comment. According to the reviewer's suggestion, we performed fluorescence, photocurrent, and impedance tests on AgI, TCNQ, and AgI@TCNQ composites with different TCNQ mass fractions. The results show that when TCNQ is coated on the surface of AgI, it will promote the separation of photogenerated carriers, which is mainly due to the fact that the core-shell structure of the composite can provide more charge transfer channels to accelerate the rapid separation of charge. The results also show that the TCNQ content has an important effect on the photogenerated charge separation efficiency. 1 wt.% AgI@TCNQ exhibits the optimal charge separation efficiency, which is proportional to its photocatalytic activity, further indicating that the charge separation efficiency directly affects its photocatalytic activity. However, it should be noted that the photoelectric performance TCNQ is better than AgI, but its photocatalytic activity is lower than AgI, due to the photocurrent is a long-term process and is limited by the carrier mobility, the continuous enhancement in photocurrent can be attributed to the relatively higher carrier mobility of TCNQ than AgI[1]. In addition, the detailed discussion have been added in the revised manuscript, thanks.

[1] Mo Zhang, Wenqing Yao, Yanhui Lv, Xiaojuan Bai, Yanfang Liu, Wenjun Jiang, Yongfa Zhu. Enhancement of mineralization ability of C3N4 via a lower valence position by a tetracyanoquinodimethane organic semiconductor, J. Mater. Chem. A, 2014, 2, 11432-11438.

[2] After all, the authors propose that (in scheme 1) Z scheme is more probable than the tranditional pathway. Thus, the electron@ TCNQ combines with holes@AgI. Then, the role of TCNQ as a good electron conductor is not important as stressed in the abstract. The role of TCNQ and AgI need to be redefined in the abstract again.

Response: Thank you for the useful comment. I have realized this problem, according to the reviewer's suggestion, the role of TCNQ and AgI have been redefined in the abstract and the introduction, thanks a lot.

[3] The authors argue that the coating of TCNQ layer with AgI can enhance the surface active sites in the abstract. Does the active site means  the site for the oxidation by the holes?

Response: Thanks for point this out. In the Z-scheme AgI@TCNQ core-shell system, Ag nanoparticles act as a recombination center for photogenerated carriers. Although this charge transfer method sacrifices a pair of photogenerated electron hole pairs, the electrons on the AgI conduction band have stronger reducing ability, while the holes enriched in TCNQ have stronger oxidizing ability. The photogenerated electrons and holes generate hydroxyl radicals and superoxide radicals by reacting with H2O and O2 to achieve degradation of organic pollutants. Therefore, this active site is not only the oxidation by the holes, but also the reduction by the electrons. In addition, the active site may exist in the surface contact between the core shell and the crystal plane of the catalyst, etc. Thus, after construction of the core-shell system, the active site must be increased, thanks a lot.

[4] The authors propose that the coating of TCNQ by AgI can enhance the stability of AgI, but the best sample AgI@TCNQ was 1wt%, which may expose both TCNQ and AgI to the solution for oxidation and reduction reactions. Thus, the enhanced stability of AgI@TCNQ vs. AgI can be quite limited or may be meaningless. This argument needs to be revised.

Response: We agree with the reviewer’s comment. The unique surface contact of the core-shell structure can increase the contact area between materials and provide more channels for the transfer of photo-generated charges, which are beneficial to photocatalytic performance. Overcoating of TCNQ on AgI can promote the rapid separation of photogenerated charge and improve photocatalytic activity, but the results have shown that the enhanced stability of AgI@TCNQ vs. AgI can be quite limited, so we agree with the reviewer's point, and the corresponding part including abstract, introduction, discussion and conclusion of the article has been corrected, thanks again.

[5] Some terminologies need to be revised to comply with common definitions in research communiities.

In the abstract, "degrade rate" -> "degradation rate", "degradation ability" may need to be replaced by another proper terminology.

"was still as high as 90%." -> 90% of what....

Response: Thanks for point this out, and we have revised it in the manuscript. Besides, the language has been polished by one English speaking expert in the revised version. thanks.  

We have tried our best to improve the manuscript and have made quite adequate changes in the revised version. We appreciate the Editors/Reviewers’ work earnestly, and hope that the correction will meet with approval.

Once again, thank you very much for your comments and suggestions.

Yours sincerely,

Weijia An

E-mail: anweijia@ncst.edu.cn

Round  2

Reviewer 1 Report

Please, check your paper carefully, e.g., no need to write "Xenon" with capital letter.

Author Response

Thanks again,I have  checked the paper carefully and made the corresponding modification.